# PROBE: Benchmarking Reasoning Paradigm Overfitting in Large Language Models

## Abstract

The reliability of reasoning benchmarks for Large Language Models (LLMs) is threatened by overfitting, which leads to inflated scores that misrepresent true capability. While existing benchmarks focus on surface-level perturbations, they fail to detect a more profound form of overfitting where models memorize problem-specific reasoning paradigms rather than developing generalizable and dynamic logical skills. To address this, we introduce PROBE (Paradigm-ReOriented Benchmark for overfitting Evaluation), a novel benchmark designed to systematically assess this limitation. PROBE introduces variants that force a shift in the core reasoning paradigm—such as simplification, introducing Unsolvability, or changing the fundamental solution approach—alongside conventional transformations. Our evaluation of state-of-the-art LLMs on PROBE reveals significant reasoning paradigm overfitting: while models achieve an average accuracy of 81.57% on original problems, their performance drops substantially to 63.18% on PROBE, with a striking low score of 35.08% on the most challenging Unsolvability type. Our work highlights the necessity for benchmarks that probe deeper into reasoning generalization and provides a tool for fostering more robust LLMs.

## 1 Introduction

Recent advancements in Large Language Models (LLMs) have significantly improved their reasoning capabilities, which now constitute a crucial dimension of LLM proficiency and are closely tracked by leading model developers (Cheng et al., 2025). To systematically assess these reasoning skills across various models, numerous benchmarks (Jimenez et al., 2024; Li et al., 2025; White et al., 2024) have been developed, aiming to accurately reflect the true reasoning capability of LLMs.

However, high scores on established benchmarks do not consistently translate into a superior perceived user experience, as practical usage often reveals a performance utility gap primarily attributable to overfitting (McLaughlin & Herlocker, 2004). Overfitting undermines evaluation effectiveness by producing inflated scores that misrepresent actual reasoning ability (Barkett et al., 2025). More critically, when overfitting benchmarks are used as a feedback metric during training, they may result in a degradation of actual reasoning capabilities (Flood et al., 2024). Therefore, distinguishing authentic reasoning from benchmark overfitting is imperative for accurate capability assessment(Ferrag et al., 2025).

While existing benchmarks have made some contributions to mitigating overfitting, their prevalent strategies such as generating variants through entity substitution (Giuliano & Gliozzo, 2008), numerical alteration (Zeng, 2024), or superficial paraphrasing primarily assess robustness against surface-level perturbations (Karimi et al., 2021). We argue that these methods exhibit inherent limitations: they are predominantly effective at detecting simple forms of overfitting rooted in pattern matching, but inadequate for identifying more profound overfitting at the level of reasoning pathways. This inadequacy occurs when **a model internalizes the underlying logical templates or solution patterns specific to a problem, rather than developing dynamic reasoning faculties under certain scenario**. Consequently, a model may appear robust to superficial changes while still relying on problem-specific cognitive shortcuts, leading to an inflated and misleading assessment of its true reasoning generalization.

Specifically, a model that has overfitting to a specific problem type tends to apply a rigid, predetermined reasoning strategy, failing to adapt its approach to the nuanced contextual demands of the

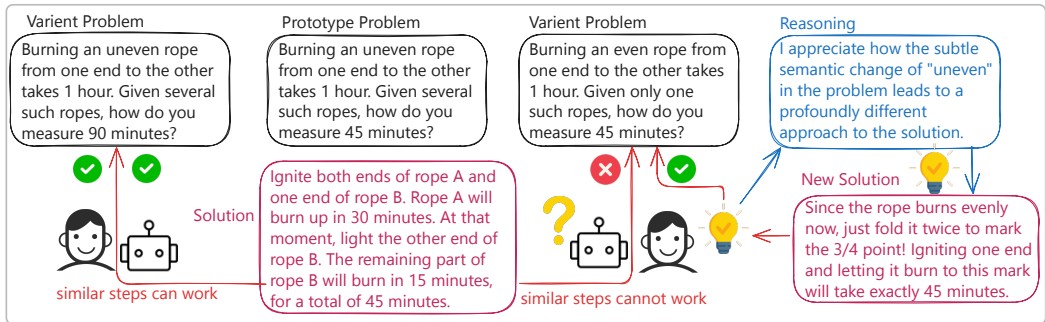

Figure 1: An illustration of problem overfitting versus true reasoning. In the "burning ropes for timing" puzzle, an overfitting model succumbs to a memorized procedure, whereas a capable reasoning agent flexibly adapts to contextual nuances.

problem instance. A representative example is the "burning ropes for timing" puzzle, as illustrated in Figure 1. The canonical solution involves igniting both ends of a rope and using two ropes for timing. However, when the problem scenario is subtly altered in a way that renders this standard paradigm less optimal or entirely inapplicable, an overfitting model is likely to persist in applying the memorized pattern rather than engaging in a new reasoning process tailored to the new constraints, unlike a truly intelligent human capable of dynamic reasoning and genuine contextual understanding. This illustrates the critical distinction between the application of a procedural memory and the flexible application of conceptual understanding and reasoning, which can hardly be evaluated by existing benchmarks.

To address this gap, we construct the **P**aradigm-**ReO**riented **B**enchmark for overfitting **E**valuation (PROBE), comprising 40 prototype problems collected from various sources with classic and fixed reasoning patterns. For each prototype, we introduce carefully designed variants to systematically evaluate reasoning overfitting from the perspective of reasoning paradigm. As shown in Table 1, our variant strategies focus on three types of reasoning pattern shifts(*Simplification*, *Unsolvability* and *Paradigm Change*) and two conventional variant types (Numerical Transformation and Paraphrasing) to allow for a more comprehensive evaluation.

We evaluate several state-of-the-art LLMs on PROBE. Our experiments reveal that mostly powerful models perform poorly, demonstrating that current LLMs suffer from significant overfitting to common reasoning paradigm and lack generalized reasoning abilities. The results show a substantial performance drop from an average accuracy of 81.57% on original problems to 63.18% on PROBE, with the most challenging *Unsolvability* type yielding only 35.08% accuracy. This performance gap is particularly striking when compared to human subjects, who achieve near-perfect scores (93.78%) on variants of this type, underscoring the limitations of current LLMs in flexibly adapting their reasoning strategies.

In summary, this work makes three key contributions: (1) It identifies and formalizes the critical limitation of existing benchmarks in detecting reasoning-level overfitting, where models memorize problem-specific solution paradigms rather than developing dynamic reasoning faculties for specific problems. (2) It introduces the PROBE benchmark, a novel evaluation framework designed to systematically assess this form of overfitting through carefully crafted variants that induce reasoning paradigm shifts. (3) It provides extensive empirical evidence demonstrating that even state-of-the-art LLMs exhibit significant vulnerability to such Paradigm Changes, highlighting a substantial gap between benchmark performance and genuine, flexible reasoning ability.

## 2 PILOT STUDY

To concretely demonstrate the concepts of our three newly defined variant forms and the phenomenon of reasoning overfitting they detect, we analyze a classic reasoning puzzle and its variants: timing with burning ropes. The canonical problem and its solution are stated as follows.

Table 1: The definitions and examples of the variant types in `PROBE`. We mark core changes in variants in red.

| Type | Definition | Example |
|---|---|---|
| Origin | Prototype problem. variants in `PROBE` are adapted from Origin. | 烧一根不均匀的绳子，从头烧到尾需要1小时，现有若干条这种绳子，如何记时45分钟？
Burning an uneven rope from one end to the other takes 1 hour.
Given several such ropes, how do you measure 45 minutes? |
| Simplification | Making a problem simpler so that it can be solved with significantly fewer original reasoning steps. | 烧一根不均匀的绳子，从头烧到尾需要1小时，现有若干条这种绳子，如何记时60分钟？
Burning an uneven rope from one end to the other takes 1 hour.
Given several such ropes, how do you measure 60 minutes? |
| Unsolvability | Altering a problem so that it cannot be solved by following the original reasoning steps. | 烧一根不均匀的绳子，从头烧到尾需要1小时，现有一条这种绳子，如何记时45分钟？
Burning an uneven rope from one end to the other takes 1 hour.
Given only one such rope, how do you measure 45 minutes? |
| Paradigm Change | Fundamentally shifting the core solving mindset of a problem, requiring a new line of thought to reach a solution. | 烧一根均匀的绳子，从头烧到尾需要1小时，现有一条这种绳子，如何记时45分钟？
Burning an even rope from one end to the other takes 1 hour.
Given only one such rope, how do you measure 45 minutes? |
| Numerical Transformation | Modifying numerical values within a problem without affecting the original reasoning path. | 烧一根不均匀的绳子，从头烧到尾需要1小时，现有若干条这种绳子，如何记时90分钟？
Burning an uneven rope from one end to the other takes 1 hour.
Given several such ropes, how do you measure 90 minutes? |
| Paraphrasing | Rewriting the problem on a semantic level without altering the original logical steps required to solve it. | 烧一根不均匀的绳子，从头烧到尾需要1小时，现在想要记时45分钟，至少需要几根绳子，怎么计时？
Burning an uneven rope from one end to the other takes 1 hour. If you want to measure 45 minutes,
what is the minimum number of ropes needed and how do you do it? |

- *Burning an uneven rope from one end to the other takes 1 hour. Given several such ropes, how do you measure 45 minutes?.*

- *Ignite both ends of rope A and one end of rope B. Rope A will burn up in 30 minutes. At that moment, light the other end of rope B. The remaining part of rope B will burn in 15 minutes, for a total of 45 minutes..*

This problem embodies a specific reasoning paradigm. We now examine how three carefully designed variants expose different aspects of reasoning overfitting.

The first variant represents a **Simplification**: *"Burning an uneven rope from one end to the other takes 1 hour. Given several such ropes, how do you measure 60 minutes?"*. In this scenario, the complex two-rope and multi-end-burning strategy is unnecessary. A model overfitting to the original paradigm might rigidly attempt to apply the multi-rope logic, failing to recognize that the goal can be achieved more simply with a single rope, thereby revealing its inflexibility.

The second variant represents a **Unsolvability**: *"Burning an uneven rope from one end to the other takes 1 hour. Given only one such rope, how do you measure 45 minutes?"*. Given that there is only one rope left, it is impossible to time with two ropes as the origin strategy does. If a model overfits to the solution pattern but fails to grasp the fundamental constraint of the number of ropes, it is likely to propose an incorrect method similar to the origin solution.

The third variant represents a **Paradigm Change**: *"Burning an even rope from one end to the other takes 1 hour. Given only one such rope, how do you measure 45 minutes?"*. The critical change from an *uneven* to an *even* rope fundamentally alters the solution pathway. An overfitting model might incorrectly apply the original solution by lighting both ends, which would take exactly 30 minutes, thus failing to measure 45 minutes. This variant reveals the ability to adapt to the new principle that an even rope can be used to measure time based on the proportion of its length burned, which in turn leads to the logic for finding the midpoint and quarter points for timing.

## 3 THE PROBE BENCHMARK

The construction of the `PROBE` is a meticulously designed, multi-stage process comprising data collection and annotation. Throughout this process, multiple annotators are involved to ensure the quality of the dataset.

### 3.1 DATA COLLECTION

The foundation of the `PROBE` is a curated set of prototype problems, each possessing a well-defined and classic reasoning pattern. To construct this set, we first collect over 100 problems from a variety of sources, including some puzzle repositories, logic puzzle websites, and Chinese social platform and then filter these suitable for generating meaningful variants with 40 prototype problems left. The list of these 40 prototype problems can be found in Appendix A.

## 3.2 DATA ANNOTATION

The annotation process for generating variants from the 40 prototype problems was conducted in three distinct stages to ensure both the diversity and quality of the resulting benchmark. The entire workflow is designed to systematically create variants that are semantically meaningful and faithfully align with our defined taxonomy.

**Stage 1: Independent Variant Generation.** In the first stage, three annotators independently generate variant questions and their corresponding answers for each prototype problem. The goal is to create a diverse pool of potential variants for each of the three core reasoning-shift types (Simplification, Unsolvability, Paradigm Change), as well as the two conventional types (Numerical Transformation, Paraphrasing). To enhance creativity and coverage, annotators are allowed to utilize large language models as an assistive tool for brainstorming potential scenario alterations (detailed in Appendix B). However, all generated variants or solutions are required to be meticulously verified and curated by the annotators.

**Stage 2: Integration and Screening.** The second stage involves another two annotators independently integrating the variant pools generated in Stage 1. The primary criterion during this screening phase is the *semantic meaningfulness* of the variant problems. Variants that are semantically inconsistent or self-contradictory are discarded. Additionally, if a prototype problem can be legitimately adapted into multiple variants of the same type, all valid instances are retained to enrich the benchmark's breadth. Conversely, if a prototype problem is inherently unsuitable for a particular variant type (e.g. a problem cannot be reasonably altered to become unsolvable), that specific variant type is simply omitted for that problem, rather than forcing a low-quality adaptation.

**Stage 3: Consistency Checking and Finalization.**

In the final stage, which is designed to ensure the reliability of the benchmark, the two sets of variants that are independently integrated during Stage 2 are systematically compared to evaluate the Inter-Annotator Agreement (IAA) (Yang et al., 2023). This consistency assessment rigorously examines both the selection of variants and the accuracy of their taxonomic classification. Since the inter-annotator consistency rate exceeds 95%, the number of discrepancies is minimal. Moreover, all instances where the annotators disagree on either the inclusion of a variant or its specific categorization undergo thorough discussion to reach a final decision. These inconsistent items after discussion are discarded to establish a unified and dependable benchmark. This stringent procedure results in a final, high-quality set of 216 variant questions, which comprises 50 variants of type *Simplification*, 45 of type *Unsolvability*, 53 of type *Paradigm Change*, 34 of type *Numerical Transformation*, and 34 of type *Paraphrasing*.

## 4 EXPERIMENTS

### 4.1 EXPERIMENTAL SETTINGS

To systematically evaluate the phenomenon of reasoning overfitting in LLMs, we conduct a comprehensive assessment using the PROBE benchmark. Our evaluation covers a diverse set of contemporary and open-sourced LLMs, including many state-of-the-art flagship models. The models examined are as follows: GPT-5(Leon, 2025), GPT-5-Mini(Leon, 2025), GPT-OSS-120B(Agarwal et al., 2025), DeepSeek-R1(Guo et al., 2025), Gemini-2.5-Pro(Huang & Yang, 2025), GPT-OSS-20B(Agarwal et al., 2025), Doubao-Seed, Qwen3-235B-A22B(Yang et al., 2025), GLM-4.5(Zeng et al., 2025), Claude-Sonnet(Choi et al., 2025), Gemini-2.5-Flash(Huang & Yang, 2025), Qwen3-235B(Yang et al., 2025), Kimi-K2(Team et al., 2025), Claude-Opus(Choi et al., 2025), GPT-4.1(Achiam et al., 2023), GLM-4.5-Air(Zeng et al., 2025), GPT-4o(Hurst et al., 2024), and GLM-4-Plus(GLM et al., 2024). To enhance evaluation consistency and minimize stochasticity, the inference parameters are fixed at a temperature of 0 and a maximum token limit of 16,384, which is sufficient for the response lengths required in our problem contexts. The correctness of each model's output is assessed automatically using GPT-4.1 as an adjudicator, which is instructed to evaluate the response against annotated reference answers from the dataset(Stephan et al., 2024).

To evaluate human performance on PROBE, two annotators who are not involved in the original dataset annotation are assigned to solve the problems. To minimize potential errors caused by oversight, the two annotators are instructed to discuss and cross-check their solutions before finalizing

Table 2: Automatic Evaluation Results on PROBE

| Model | Origin | V1 | V2 | V3 | V4 | V5 | Average | Rank |
|---|---|---|---|---|---|---|---|---|
| GPT-5 | 88.89 | 82.00 | 71.43 | 88.24 | 90.91 | 94.12 | 84.76 | **2** |
| GPT-5-Mini | 90.00 | 81.63 | 55.56 | 83.02 | 96.97 | 91.18 | 80.37 | **3** |
| GPT-OSS-120B | 83.78 | 70.83 | 53.33 | 67.35 | 88.24 | 87.50 | 71.63 | 4 |
| DeepSeek-R1 | 81.58 | 76.00 | 43.18 | 75.47 | 76.47 | 84.85 | 70.56 | 5 |
| Gemini-2.5-Pro | 87.50 | 72.00 | 35.56 | 71.70 | 91.18 | 91.18 | 70.37 | 6 |
| GPT-OSS-20B | 84.21 | 69.39 | 47.62 | 73.08 | 73.53 | 81.25 | 68.42 | 7 |
| Doubao-Seed1.6 | 90.00 | 68.75 | 31.11 | 62.26 | 93.94 | 94.12 | 67.14 | 8 |
| Qwen3-235B-A22B | 80.00 | 70.00 | 44.44 | 66.04 | 73.53 | 85.29 | 66.67 | 9 |
| GLM-4.5 | 82.05 | 74.00 | 34.15 | 66.04 | 75.76 | 82.35 | 65.88 | 10 |
| Claude-Sonnet | 77.50 | 64.00 | 28.89 | 71.70 | 76.47 | 81.82 | 63.26 | 11 |
| Gemini-2.5-Flash | 82.50 | 66.00 | 24.44 | 56.60 | 85.29 | 88.24 | 61.57 | 12 |
| Qwen3-235B | 85.00 | 60.00 | 15.56 | 66.04 | 90.91 | 85.29 | 60.93 | 13 |
| Kimi-K2 | 82.50 | 70.00 | 28.89 | 58.49 | 76.47 | 73.53 | 60.19 | 14 |
| Claude-Opus | 82.50 | 72.00 | 28.89 | 50.94 | 70.59 | 79.41 | 58.80 | 15 |
| GPT-4.1 | 78.95 | 65.31 | 52.27 | 43.14 | 67.65 | 69.70 | 58.29 | 16 |
| GLM-4.5-Air | 76.32 | 60.00 | 25.00 | 56.60 | 74.19 | 73.53 | 56.13 | 17 |
| GPT-4o | 70.00 | 38.00 | 8.89 | 35.85 | 50.00 | 70.59 | 38.43 | 18 |
| GLM-4-Plus | 65.00 | 32.00 | 2.22 | 30.19 | 50.00 | 67.65 | 33.80 | 19 |
| **Overall** | **81.57** | **66.22** | **35.08** | **62.38** | **77.89** | **82.31** | **63.18** | - |
| **Human** | - | 100.00 | 97.78 | 92.16 | 85.29 | 90.91 | 93.87 | **1** |

**Note:** The **Overall** score for each type (Origin and 5 variants) is the mean across all models for that specific type. The **Average** score for each model is the mean of its performance across all five variant types (excluding Origin). The **Rank** represents ranking based on Average score. **V1-V5** represents Simplification, Unsolvability, Paradigm Change, Numerical Transformation and Paraphrasing respectively. To facilitate comparison, the top three performers in each category are highlighted in red, green, and yellow respectively.

a consensus answer. It is important to note that this evaluation aims to assess the human ability to transfer reasoning patterns to variant problems(Zhang et al., 2024b). Under this setup, participants are first presented with the original question and its reference solution to familiarize them with the original reasoning approach, and are then asked to solve its variants.

## 4.2 MAIN RESULTS

The main results are presented in Table 2 and we have the following insights:

**The prevalence and severity of reasoning paradigm overfitting are striking.** A significant performance degradation is observed across all flagship models on variant problems, with the average accuracy dropping substantially from 81.57% on original problems to 63.18% on PROBE. This decline is particularly pronounced on the first three variant types designed to challenge reasoning paradigms, unequivocally demonstrating that current LLMs suffer from severe overfitting to specific reasoning templates rather than possessing generalizable reasoning abilities.

**LLMs exhibit good robustness to superficial perturbations without paradigm shifting.** In stark contrast to reasoning paradigm overfitting, performance on the latter two variant types — Numerical Transformation (77.89% accuracy) and Paraphrasing (82.31% accuracy) shows similar performance compared to original problems, with Paraphrasing even slightly outperforming the original problems. This clear dichotomy indicates that flagship models have largely overcome simple pattern matching and can robustly apply the same reasoning paradigm across diverse scenarios to solve problems.

**A significant performance gap exists between model structures and series, with GPT-5 demonstrating notable advantages.** The GPT-5 series models (GPT-5 and GPT-5-mini) achieve the highest overall accuracy, significantly outperforming subsequent models. The substantial improve-

Table 3: Meta evaluation results. We demonstrate scores when different judge models (and human evaluator) assess human-generated answers across five variant types and calculate the correlation between different judge models and human evaluator.

| Judge | V1 | V2 | V3 | V4 | V5 | Average | Correlation |
|---|---|---|---|---|---|---|---|
| GPT-4.1 | 100.00 | 97.78 | 92.16 | 85.29 | 90.91 | 93.87 | 0.7882 |
| GPT-5 | 98.00 | 97.78 | 90.57 | 79.41 | 91.18 | 92.13 | 0.7028 |
| Gemini-2.5-Pro | 98.00 | 97.78 | 84.91 | 88.24 | 91.18 | 92.13 | 0.7028 |
| GLM-4.5 | 98.00 | 95.56 | 84.91 | 82.35 | 88.24 | 90.28 | 0.5337 |
| DeepSeek-v3 | 82.00 | 46.67 | 66.04 | 73.53 | 82.35 | 69.44 | 0.2836 |
| Human | 100.00 | 97.78 | 88.24 | 88.24 | 96.97 | 94.34 | - |

**Note: V1-V5** represents Simplification, Unsolvability, Paradigm Change, Numerical Transformation and Paraphrasing respectively. The **Average** score for each model is the mean of its performance across all five variant types. The reported correlations are Spearman's rank correlation coefficients with human judgment.

ment over the poorer-performing GPT-4o generation suggests that during the evolution to GPT-5, its developers have recognized this type of reasoning overfitting and taken useful methods to mitigate it. Conversely, other flagship models like Doubao-Seed1.6, while achieving competitive performance on original problems and the last two types variants (Numerical Transformation and Paraphrasing, without paradigm shifting), exhibit dramatic performance drops on the first three variants, indicating that overcoming reasoning paradigm overfitting remains a major challenge for most model developers.

**Among reasoning paradigm shifts, Unsolvability variant presents the greatest challenge.** The performance on Unsolvability variants (35.08% accuracy) shows the most severe decline, significantly lower than other reasoning-shift categories. This indicates that models struggle profoundly when problem conditions change to render standard solutions inapplicable, often failing to recognize the fundamental constraint shifts and persisting in applying memorized but now invalid reasoning patterns.

**Human performance demonstrates remarkable robustness and generalization across all variant types, sharply contrasting with model limitations.** As shown in the final row of Table 2, human subjects achieve near-perfect accuracy on the first three reasoning paradigm variants (Simplification, Unsolvability and Paradigm Change), with scores of 100.00%, 97.78%, and 92.16% respectively. This exceptional performance underscores humans' inherent ability to flexibly adapt their reasoning strategies to fundamental problem structure changes, without being constrained by specific solution templates. Interestingly, for the latter two variant types, several state-of-the-art (SOTA) models match or even surpass human performance. This suggests that current models have significantly reduced their overreliance on superficial pattern matching. This phenomenon further validates the significance of our proposed first three variant types, which are designed to assess overfitting based on shifts in reasoning paradigms.

## 4.3 META EVALUATION

To ensure the reliability of our automated evaluation, we conducted a meta-evaluation comparing the judgment capabilities of several LLMs against human evaluators(Yu et al., 2025). As shown in Table 3, we measured each model's performance as a judge based on its average agreement score when evaluating human responses across all problem types, along with its Spearman's rank correlation coefficient with human ratings. The results reveal significant differences among the models. GPT-4 achieved the highest average score (93.87%) and the strongest correlation with human judgment (Spearman's $\rho$=0.7882), demonstrating its superior capability as a reliable evaluator. Models such as GPT-5, Gemini 2.5 Pro, and GLM-4.5 showed competitive but slightly lower performance, while DeepSeek-V3 lagged considerably. Based on these findings, we selected GPT-4 as the judge model for all subsequent experiments in our study, as its evaluations most closely approximate those of human assessors.

## 4.4 PROMPT-BASED INTERVENTIONS

Building upon findings that reveal significant reasoning paradigm overfitting in LLMs, we further investigate whether carefully designed prompting strategies can mitigate this issue. We evaluate four distinct prompting approaches using six representative models (each exhibiting varying performance levels on PROBE) to assess their effectiveness in enhancing model adaptability to reasoning paradigm shifts.

---

请仔细阅读以下问题并解答，注意它可能包含与经典问题不同的关键条件，请谨慎套用熟悉的解法。
[问题]：
Please read the following question carefully and answer it, noting that it may contain key conditions that differ from classical problems. Be cautious about applying familiar solutions directly.
[Question]：    # Simple Warning

请你扮演一个严谨的逻辑学家，你的特点是从不盲目套用公式或既定策略，而是从第一性原理出发分析问题。现在请解决以下问题：
[问题]：
Please act as a rigorous logician. Your characteristic is to never blindly apply formulas or predetermined strategies, but rather to analyze problems from first principles. Now, please solve the following question:
[Question]：    # Role-Playing

请仔细阅读以下问题并解答，按以下步骤思考：**步骤一：分析差异。** 先将此问题与你熟悉的类似经典问题进行比较，找出核心条件上的关键差异。**步骤二：评估可行性。** 基于这些差异，判断经典解法是否仍然完全适用、部分适用、或不适用。**步骤三：制定新方案。** 如果经典解法不适用，请构思一个新的解决方案。
[问题]：
Please read the following question carefully and answer it by following these steps:
Step 1: Analyze the differences.
Begin by comparing this problem with similar classic problems you are familiar with, and identify the key differences in the core conditions.
Step 2: Assess feasibility.
Based on these differences, determine whether the classic solution is still fully applicable, partially applicable, or not applicable at all.
Step 3: Develop a new plan.
If the classic solution is not applicable, devise a new solution.
[Question]：    # Meta-Cognitive Prompting(Chain-of-Thought)

---

Figure 2: Under the three strategies of Simple Warning, Meta-Cognitive Prompting, and Role-Playing, the specific prompts we used (StraightForward does not require an additional prompt).

The four prompting approaches are defined as follows and detailed in Figure 2.

- **StraightForward:** The baseline approach with no additional instructions.
- **Simple Warning:** A minimal intervention that alerts the model to potential differences from classic problems.
- **Meta-Cognitive Prompting:** A structured Chain-of-Thought approach that explicitly guides the model through comparative analysis and solution adaptation.
- **Role-Playing:** An approach that frames the task within a specific cognitive persona to encourage principled reasoning.

Table 4: Models performance on PROBE and original problems with different prompt strategies.

|  | Prompt Strategy | Claude | Doubao | GLM-4.5 | GPT-4.1 | GPT-4o | Kimi-K2 |
|---|---|---|---|---|---|---|---|
| **PROBE** | StraightForward | 63.26 | 67.14 | 65.88 | 58.29 | 38.43 | 60.19 |
|  | Simple Warning | **66.82** | **73.15** | 67.51 | 68.06 | **40.28** | **64.81** |
|  | Meta Cognitive | 59.07 | 72.90 | 66.83 | 65.28 | 31.94 | 53.24 |
|  | Role Playing | 63.89 | 67.13 | **71.05** | **71.76** | 35.19 | 62.50 |
| **Origin** | StraightForward | **77.5** | 90 | 82.05 | 78.95 | **70** | **82.5** |
|  | Simple Warning | **77.5** | **95** | **83.78** | 70 | 62.5 | 80 |
|  | Meta Cognitive | 70 | 90 | 82.05 | 57.5 | 52.5 | 67.5 |
|  | Role Playing | **77.5** | 92.5 | 74.29 | **80** | 60 | 72.5 |

**Note:** Claude and Doubao stand for Claude-Sonnet and Doubao-Seed1.6 respectively. The best results across different strategies are shown in **bold**.

On the evaluation of different prompt strategies, it is essential to assess not only their performance on the PROBE dataset but also whether they cause performance degradation on the original problems(Mai et al., 2025). Therefore, we compare the effects of each strategy on both PROBE and the

Table 5: Model Performance Comparison with Different Prompting Strategies

| Default | Model | Origin | V1 | V2 | V3 | V4 | V5 | Avg. |
|---|---|---|---|---|---|---|---|---|
| **StraightForward** | | | | | | | | |
| | Claude-Sonnet | 77.50 | 64.00 | 28.89 | 71.70 | 76.47 | 81.82 | 63.26 |
| | Doubao-Seed1.6 | 90.00 | 68.75 | 31.11 | 62.26 | 93.94 | 94.12 | 67.14 |
| | GLM-4.5 | 82.05 | 74.00 | 34.15 | 66.04 | 75.76 | 82.35 | 65.88 |
| | GPT-4.1 | 78.95 | 65.31 | 52.27 | 43.14 | 67.65 | 69.70 | 58.29 |
| | GPT-4o | 70.00 | 38.00 | 8.89 | 35.85 | 50.00 | 70.59 | 38.43 |
| | Kimi-K2 | 82.50 | 70.00 | 28.89 | 58.49 | 76.47 | 73.53 | 60.19 |
| **Simple Warning** | | | | | | | | |
| | Claude-Sonnet | 77.50 | 74.00 | 40.91 | 75.47 | 70.59 | 72.73 | 66.82 |
| | Doubao-Seed1.6 | 95.00 | 84.00 | 24.44 | 81.13 | 91.18 | 91.18 | 73.15 |
| | GLM-4.5 | 83.78 | 70.21 | 56.76 | 62.00 | 78.79 | 73.33 | 67.51 |
| | GPT-4.1 | 70.00 | 76.00 | 64.44 | 64.15 | 67.65 | 67.65 | 68.06 |
| | GPT-4o | 62.50 | 44.00 | 13.33 | 39.62 | 47.06 | 64.71 | 40.28 |
| | Kimi-K2 | 80.00 | 74.00 | 40.00 | 56.60 | 79.41 | 82.35 | 64.81 |
| **Meta Cognitive** | | | | | | | | |
| | Claude-Sonnet | 70.00 | 62.00 | 28.89 | 66.04 | 69.70 | 73.53 | 59.07 |
| | Doubao-Seed1.6 | 90.00 | 82.00 | 32.56 | 77.36 | 88.24 | 88.24 | 72.90 |
| | GLM-4.5 | 82.05 | 66.00 | 50.00 | 73.58 | 70.00 | 75.76 | 66.83 |
| | GPT-4.1 | 57.50 | 70.00 | 71.11 | 58.49 | 58.82 | 67.65 | 65.28 |
| | GPT-4o | 52.50 | 42.00 | 4.44 | 28.30 | 41.18 | 50.00 | 31.94 |
| | Kimi-K2 | 67.50 | 54.00 | 40.00 | 54.72 | 58.82 | 61.76 | 53.24 |
| **Role Playing** | | | | | | | | |
| | Claude-Sonnet | 77.50 | 72.00 | 24.44 | 62.26 | 85.29 | 85.29 | 63.89 |
| | Doubao-Seed1.6 | 92.50 | 74.00 | 22.22 | 71.70 | 85.29 | 91.18 | 67.13 |
| | GLM-4.5 | 74.29 | 74.47 | 55.56 | 67.35 | 85.19 | 77.42 | 71.05 |
| | GPT-4.1 | 80.00 | 76.00 | 66.67 | 62.26 | 79.41 | 79.41 | 71.76 |
| | GPT-4o | 60.00 | 34.00 | 8.89 | 28.30 | 50.00 | 67.65 | 35.19 |
| | Kimi-K2 | 72.50 | 72.00 | 42.22 | 58.49 | 70.59 | 73.53 | 62.50 |

**Note:** V1=Simplification, V2=Unsolvability, V3=Paradigm Change, V4=Numerical Transformation, V5=Paraphrasing. The **Avg.** score for each model is the mean of its performance across all five variant types.

original task as depicted in Table 4. (For `PROBE`, we only present the average score in Table 4, detailed results are provided in Table 5)

Our experiment reveal several key findings: First, regardless of the prompting approach, models consistently perform much worse on PROBE than on the original problems, indicating that reasoning paradigm overfitting cannot be easily resolved through simple prompt engineering. Second, different prompt strategies do influence performance on both PROBE and the original questions. Overall, Simple Warning proves most effective—it not only improves performance on PROBE for most models but also incurs minimal performance drop on the original task. In contrast, while Meta-Cognitive Prompting and Role-Playing occasionally lead to notable gains on PROBE for certain models (e.g., Role-Playing helps GLM-4.5 and GPT-4.1 achieve the highest PROBE scores), they often result in significant degradation on the original problems.

In summary, although reasoning paradigm overfitting remains a persistent challenge, it can be partially mitigated through lightweight interventions such as Simple Warning. This offers practical insight for real-world applications involving similar scenario.

### 4.5 CASE STUDY

Noting that V2 problems (Unsolvability) exhibit a particularly low performance of all variants, we select for detailed analysis a problem that has two Type II variants: the Round Table Coin Placement problem.

The first varient problem describes *a round table with a coin already at the center; two players take turns placing non-overlapping coins, and the player who cannot place a coin loses*. The second varient problem change the round table to a multi-round one.

This alteration invalidates the classic "center-symmetry" strategy, which is only applicable to perfectly symmetrical shapes with no coin in the table center.

In these two cases, models often overlook the initial condition that the center is already occupied, leading them to suggest an invalid first move of "occupying the center and they fail to recognize that a semi-circular table breaks the central symmetry, making the core "mirroring" strategy inapplicable. Instead of adapting their reasoning to the new geometric constraints, models persistently apply the memorized paradigm, resulting in a logically structured but fundamentally flawed answer. There is a typical erroneous response (simplified version, whole response can been seen in Appendix C):

> **Typical Erroneous Response:** "As the first player, you have a winning strategy. First, place a coin at the center of the table. Then, whenever your opponent places a coin at any point P, you immediately place a coin at the point P' that is symmet-

rically opposite with respect to the center. This mirroring strategy guarantees that you always have a move, ensuring your victory."

## 5 Related Work

### 5.1 Evaluation of Large Language Models

The evaluation of reasoning capabilities in large language models has become a central focus of NLP research, driven by the rapid progression of model scale and architectural innovations(Wang, 2024). Numerous benchmarks have been established to quantitatively assess these capabilities, spanning domains such as mathematical reasoning(Mishra et al., 2022), commonsense reasoning(Davis, 2023), and complex problem-solving(Zhang et al., 2025). These benchmarks aim to provide standardized measures of abstract reasoning(Lu et al., 2021), logical deduction(Luo et al., 2023), and multi-step inference(Fujisawa et al., 2024). While reported scores on these benchmarks have consistently risen, reflecting apparent improvements in model sophistication, concerns have grown regarding the extent to which these metrics genuinely capture broad, generalizable reasoning skills versus the ability to exploit statistical patterns within benchmark datasets(Banerjee et al., 2024). This has prompted a critical line of inquiry into the robustness and true generalization of the reasoning processes these models employ.

### 5.2 Overfitting in Reasoning

The challenge of overfitting plagues the evaluation of reasoning capabilities in large language models. While numerous benchmarks Li et al. (2024); Mirzadeh et al. (2024) have been developed to assess mathematical reasoning skills, their effectiveness is undermined when models achieve high scores through memorization of solution patterns rather than genuine reasoning ability. Existing approaches to mitigate this issue typically rely on surface-level perturbations such as entity substitution, numerical alteration, or paraphrasing. However, these methods primarily test robustness against lexical and syntactic variations, failing to address a more profound form of overfitting where models internalize the underlying logical templates of specific problem types.

## 6 Limitation

To ensure high-quality variants that effectively probe reasoning patterns, we meticulously collect prototype problems with distinct reasoning paradigms from publicly available sources and employed manual annotation. While this process guarantees the benchmark's quality and conceptual rigor, it necessarily restricts its size due to the significant resource costs associated with detailed annotation(Zhang et al., 2024a; Villalobos et al., 2024). Despite this limitation in scale, the core contribution of our work transcends the benchmark itself. The proposed framework of creating variants based on reasoning pattern shifts (Simplification, Unsolvability and Paradigm Change) provides a generalizable and impactful methodology for assessing deep reasoning overfitting beyond surface-level perturbations. This conceptual approach can be productively applied to other problem domains and future benchmarks to evaluate model robustness more profoundly(Jeppsson & Pons, 2004). We anticipate that subsequent research within the community will build upon this paradigm of reasoning-centric evaluation, extending it to larger datasets and diverse reasoning tasks to further advance the development of genuinely robust language models(Hassid et al., 2024).

## 7 Conclusion

In conclusion, our study reveals a critical disconnect between the benchmark performance and genuine reasoning ability of large language models (LLMs). Through the introduction of PROBE, a benchmark designed to test for reasoning paradigm overfitting, we demonstrate that even state-of-the-art models exhibit significant rigidity, failing to adapt when classic solution paths are invalidated or altered. The stark contrast between this model vulnerability and near-perfect human performance underscores that current LLMs often rely on memorized procedures rather than flexible, generalizable reasoning. These findings highlight the necessity of moving beyond static benchmarks to foster the development of more robust and truly intelligent AI systems.

## 8 ETHICS STATEMENT

This work investigates the phenomenon of reasoning paradigm overfitting in large language models (LLMs). Our experiments are built upon the proposed PROBE benchmark, which is constructed from publicly available classic reasoning puzzles. Critically, all prototype problems, their variants, and corresponding gold-standard solutions were meticulously annotated and verified by human experts to ensure semantic validity and correctness. We confirm that no private data or personally identifiable information (PII) was involved in this research. The proposed PROBE benchmark is designed to expose a critical limitation in current LLM evaluation practices, thereby benefiting the research community by enabling a more accurate assessment of genuine, generalizable reasoning abilities. While overfitting to benchmarks may remain an inherent challenge in LLM development, our benchmark serves as a crucial diagnostic tool for fostering more robust and trustworthy reasoning models. We affirm that our research fully complies with the ACL Code of Ethics and poses no foreseeable harm to individuals or groups.

## 9 REPRODUCIBILITY STATEMENT

To ensure the reproducibility and transparency of our results, all 216 varient problems in `PROBE` and 40 original problems and evaluation scripts have been submitted as supplementary materials. These materials include detailed instructions for readers to reproduce the experiments reported in this paper.

## 10 THE USE OF LARGE LANGUAGE MODELS (LLMS)

LLMs were utilized to support the writing process of this paper. Specifically, they provided assistance with grammar correction, wording refinement, and formatting adjustments. Furthermore, we use LLM as a assistant in the process of data construction that have been detailed in the manuscript. We affirm that the use of AI tools does not affect the originality of this work, and the authors remain fully responsible for the content and accuracy of the paper.

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

Table 6: List of 40 classic reasoning problems used as prototype in PROBE

| | |
|---|---|
| 1. 烧绳计时 Burning Ropes for Timing | 2. 水桶量水 Water Jug |
| 3. 天平找重球 Finding Heavy Ball with Balance | 4. 三门 Monty Hall |
| 5. 盲人分牌 Blind Card Division | 6. 天平砝码分盐 Dividing Salt with Balance and Weights |
| 7. 拿苹果 Taking Apples | 8. 盲人分袜子 Blind Sock Matching |
| 9. 猴子搬香蕉 Monkey Moving Bananas | 10. 飞机绕地球 Airplane Around the World |
| 11. 弹球概率 Ball Bouncing Probability | 12. 切金条 Cutting Gold Bar |
| 13. 硬币公平 Fair Coin | 14. 沙漏计时 Hourglass Timing |
| 15. 四人过桥 Four People Crossing a Bridge | 16. 拿钻石 Taking Diamonds |
| 17. 盒子标签 Box Labeling | 18. 苹果装箱 Packing Apples |
| 19. 圆桌放硬币 Placing Coins on a Round Table | 20. 空瓶换饮料 Empty Bottles for Drinks |
| 21. 赛马 Horse Racing | 22. 找变质药 Finding Expired Medicine |
| 23. 囚犯释放概率 Prisoner Release Probability | 24. 硬币摆放 Coin Arrangement |
| 25. 分水 Water Division | 26. 真假话问路 Truth-teller and Liar Path |
| 27. 海盗分金币 Pirate Gold Division | 28. 扔鸡蛋 Egg Dropping |
| 29. 农夫过河 Farmer Crossing River | 30. 商人随从过河 Merchants and Servants Crossing River |
| 31. 抓狐狸 Catching a Fox | 32. 灯泡开关 Light Bulb Switching |
| 33. 找好芯片 Finding Good Chip | 34. 鳄鱼吃猴子 Crocodile Eating Monkey |
| 35. 性别概率 Gender Probability | 36. 夫妇握手 Couples Handshaking |
| 37. 村子女人数量 Village Women Count | 38. 考试及格人数 Exam Passers Count |
| 39. 病狗 Sick Dog | 40. 囚犯抓绿豆 Prisoners and Green Beans |

## A   ALL PROTOTYPE PROBLEMS IN PROBE

All prototype problems in PROBE are classic strategic reasoning problems, as listed in Table 6.

## B   PROMPT FOR GENERATING VARIANTS IN PROBE

The prompt we design for the annotators to call LLMs is shown in the figure 3. A key point to highlight is that the use of LLMs is strictly assistive. Annotators are instructed not to accept the LLMs' response verbatim. Instead, the process is designed to leverage the model's generative capability for idea inspiration only. The core responsibilities of critical evaluation, factual verification, linguistic refinement, and final curation remain entirely with the human annotators.

## C   DETAILED ERRONEOUS RESPONSE IN CASE STUDY

The whole response of the Placing Coins on a Round Table problem is as follows(generated by Doubao-Seed). It can be easily seen that this response adheres to the reasoning paradigm of the original problem, ignoring the detailed changes in the new problem that lead to Unsolvability.

要解决正方桌硬币放置游戏的先手必胜问题，**核心策略是利用"中心对称"性质**，通过对称操作确保每一步后手有位置时，先手必然有对应位置，最终迫使后手无法放置硬币。**1. 游戏规则与关键性质** - 桌子是**中心对称图形**（正方形的中心是对角线交点，也是对称中心）。- 硬币不能重叠，不能超出桌子边界。- 无法放置硬币的一方输。**2. 先手必胜策略** **第一步：先手将第一枚硬币放在桌子的** **正中心**（对称中心）。**后续步骤：每当后手在某个位置放置一枚硬币后，先手立即在该位置的** **中心对称点**（关于桌子中心的对称点）放置一枚硬币。**3. 策略的正确性证明** - **对称性保证合法性**：正方形是中心对称图形，若后手放置的硬币位置(P)在桌子内且不重叠，则其对称点(P')必然也在桌子内（对称性质），且不会与已有的硬币重叠（因为(P)未重叠，(P')与(P)对称，不会与(P)或其他对称点重叠）。- **必胜性保证**：先手第一步占据中心后，每一步后手的操作都能被先手的对称操作抵消。只要后手有位置放硬币，先手就一定有对应的对称位置放硬币。最终，**后手会先遇到没有位置可放的情况**，先手获胜。**总结** 先手的必胜策略是：**第一步放中心，之后每一步都放后手位置的中心对称点**。该策略利用中心对称性质，将"是否有位置"的问题转移给后手，确保先手始终有应

你是一个人类出题专家，希望考核你的学生对于一些策略推理问题是不是真的完全理解了还是只是'记住了'解题步骤。为此，你需要对一些策略问题做改编，提高对学生的考验难度。

改编思路有很多种。例如你可以将问题中的数字改得简单，让问题只需要更少的步骤就能解决，考验学生是不是还像原来那样用原来的复杂步骤解题；
你也可以通过某种方式修改问题，让问题变得不可解，原有思路实效，考验学生是不是能够意识到这一点；
你还可以将问题中的某个条件作修改、删除，或新增某种限制，让解题思路完全变化，考验学生是不是能够理解不同情景的变化会导致原有推理范式不可复用；
针对特定问题如果你有更好的改编思路，鼓励你做更多有趣的改编。我会给你一个案例作为参考。

【案例】
<origin_question> 烧一根不均匀的绳子，从头烧到尾需要1小时，现有若干条这种绳子，如何记时45分钟？

<new_question_1> 烧一根不均匀的绳子，从头烧到尾需要1小时，现有若干条这种绳子，如何记时60分钟？
<reason_1> 把45分钟改成60分钟，问题变得十分简单，考验学生是否能够意识到只需要用一小步就能够完成任务。

<new_question_2> 烧一根不均匀的绳子，从头烧到尾需要1小时，现有一条这种绳子，如何记时45分钟？
<reason_2> 绳子数量变成只有一根，这样就没办法像原本那样用多根绳子来计时了，导致原有思路实效。

<new_question_3> 烧一根均匀的绳子，从头烧到尾需要1小时，现有一条这种绳子，如何记时45分钟？
<reason_3> 绳子数量变成只有一根，但是绳子变成均匀的了，这样就会有按照烧的位置记时的新方案。

下边是需要你来改编的原问题，请你按照<new_question_i>\<reason_i>的形式来改编问题，如果你有好的思路，可以不局限于三条，但是你需要注意的是，案例中的三个改编后问题都是合理的，并且每个问题是单独的，请你不要出一些不合理的问题。

【需要你来改编的问题】

{Question}

You are a human expert in creating exam questions, aiming to assess whether your students have truly fully understood certain strategic reasoning problems, or if they have merely "memorized" the solution steps. To achieve this, you need to adapt some strategic problems to increase the difficulty of the assessment for students.

There are many adaptation approaches.For example, you can simplify the numbers in the problem so that it requires fewer steps to solve, testing whether the student still uses the original complex steps as before.
You can also modify the problem in a way that makes it unsolvable, rendering the original approach ineffective, to test whether the student can realize this.
Additionally, you can alter, remove, or add certain constraints to the problem, completely changing the solution approach, to test whether the student understands that changes in different scenarios make the original reasoning paradigm inapplicable.
Of course, if you have better adaptation ideas for specific problems, you are encouraged to make more interesting modifications. There is an example for reference.

【Example】
<original_question> Burning an uneven rope from start to finish takes 1 hour. Given several such ropes, how can you measure 45 minutes?

<new_question_1> Burning an uneven rope from start to finish takes 1 hour. Given several such ropes, how can you measure 60 minutes?
<reason_1> Changing 45 minutes to 60 minutes makes the problem very simple, testing whether the student can realize that only a small step is needed to complete the task.

<new_question_2> Burning an uneven rope from start to finish takes 1 hour. Given one such rope, how can you measure 45 minutes?
<reason_2> The number of ropes is reduced to only one, making it impossible to use multiple ropes for timing as in the original approach, thus invalidating the original method.

<new_question_3> Burning an even rope from start to finish takes 1 hour. Given one such rope, how can you measure 45 minutes?
<reason_3> The number of ropes is reduced to only one, but the rope becomes even, allowing for a new timing solution based on the burning position.

Below is the original problem that you need to adapt. Please present your adaptations in the format of <new_question_i> <reason_i>. If you have good ideas, you are not limited to three adaptations. However, please note that the three adapted questions in the example are all reasonable, and each question is independent. Avoid creating unreasonable problems.

【Problem for you to adapt】
{Question}.

Figure 3: Our designed prompt for calling LLM to generate variants.

对方案，最终获胜。**答案：** 先手第一步将硬币放在桌子正中心，之后每当后手放一枚硬币，先手就放在该硬币关于桌子中心的对称位置。这样先手必能获胜。

