# OpenReview forum: "PROBE: Benchmarking Reasoning Paradigm Overfitting in Large Language Models"
_ICLR.cc/2026/Conference — ICLR 2026 Conference Withdrawn Submission_

### Official Review · Reviewer_mtBV · 2025-10-28

**Soundness:** 2
**Presentation:** 2
**Contribution:** 2
**Rating:** 2
**Confidence:** 4

**Summary:**

In this paper the authors present PROBE, a benchmark to evaluate reasoning capability in LLMs, with specific focus on evaluating the ability of the models to apply reasoning in cases where the pattern of the problem deviates from well known cases.

- The benchmark takes 40 curated problem stems, and creates variants of the problem to evaluate simplification, making the problem unsolvable, paradigm change, numerical changes, and paraphrasing. This totals 216 questions.

- An extensive range of SOTA models are evaluated against this benchmark, with results presented across the whole dataset and each variant class.

- Human performance is evaluated as a baseline to compare the models against, and the authors use GPT-4.1 as an LLM as a judge to evaluate the correct answers.

- The authors also provide the template to follow to create additional questions of this form and recognize there is a limitation in the form of total dataset size.

**Strengths:**

- The benchmark attempts to address a crucial area of LLM reasoning - and quantifies the failings across several key variations - this is a complex area to define and the paper makes some good progress here

- The data collection method is provided in good detail making it an easy study to replicate and transparent to increase the dataset size.

- The analysis covers the variations in the dataset well, and presents some thoughts about why different tasks are found to have different levels of complexity.

- And evidence is shown to explain how the models are prompted / how failure modes occur.

**Weaknesses:**

- The biggest weakness the authors identify, which is the limited number of questions in the benchmark - improving this would go a long way towards improving the benchmark resolution and utility for the wider research community

- The proposed set of variants is a useful addition to the original problem, but faces the same limitation, it is well known that simple variations cause problems for models and increasing numbers of variations of traditional problems get incorporated into the training data.

- I have some concern about benchmark saturation / longevity - given the high scores achieved by GPT-5 - especially in V4/5 - I question how useful the paraphrasing / numerical variation is. Indeed the fact that quite a few of the models outperform the original form on this benchmark suggests it doesn’t provide as much of an additional challenge as expected. For this benchmark to provide a stronger set of evaluations I wonder if that authors have ideas to expand the variation set further? Or to increase the complexity of the problems?

- The framing of the paper asserts that this effect is due to overfitting to the problem structure seen often in the training data - personally I feel that this statement lacks evidence to support it, I know it’s a commonly held belief, but I  found the way this was asserted largely with quite out of date references made me question that claim rather than being convinced by it. Could you expand the motivation section to better frame the evidence that this is a genuine overfitting problem?

**Questions:**

- Given that the simple warning seems to represent a moderate performance improvement of approximately 5%, was there a reason that the main results in Table 2 are only provided with the simple prompt? (Please correct me if I’ve misunderstood these results.) My suspicion is this analysis was carried out after collecting the main results, but I wonder if the authors could comment on this / consider updating the results in the main table?

- I appreciate this is unlikely in the rebuttal period - but could the authors comment on how hard it would be to increase the total dataset size?

- Do the authors intend this benchmark to be a public leaderboard that others can evaluate against? In which case can they comment on the dataset availability and consider addressing my above comment to ensure the best results are shown with the best method.

- Or is the work intended more to prove the overfitting claim regarding problem structure? In which case I would appreciate a slightly deeper analysis / a stronger conclusion.

- I could be convinced by either argument, but currently the paper feels like it sits between the two and needs strengthening in one direction or the other.


Nitpicks / styling

- The 4.s.f in Table 2 is inappropriate given the total number of questions in the benchmark

- The styling of figures / tables with a caption and then an additional note makes the presentation feel a bit messy and cluttered. Perhaps condense the captions into a single block?

- The case study presented in 4.2 felt out of place, maybe could be moved to the appendix. It’s useful to see, but feels disconnected from the flow of the paper.

---

### Official Review · Reviewer_qDk8 · 2025-10-28

**Soundness:** 2
**Presentation:** 2
**Contribution:** 2
**Rating:** 2
**Confidence:** 5

**Summary:**

- A novel benchmark that allows to asses how language models can overfit on specific logical templates which will lead to high scores on benchmarks but not necessarily mean the model possess true reasoning capability and can generalize well.
- Data collection: 100 problems from puzzle repositories, logic puzzles etc.
- Creation of different types of problem variants: using human annotators generate new variants of each problem by using reasoning shifts (simplification, insolvability and paradigm shift), numerical transformation and paraphrasing.
- Experiments setup:
	- No sampling (temperature 0), max tokens 16384.
	- GPT-4.1 used to judge the correctness of responses by comparing them to the ground truth solutions.
- Results:
	- All of the frontier models fail the most on V2 and V3 (insolvability, paradigm change).
	- Humans can solve the problems and the variations. However, on paraphrasing and numerical transformations humans actually have worse performance some LLMs, e.g. GPT-5 and GPT-5-mini.
	- Meta-evaluation: compared how accuracies asses by different models GPT-4.1, GPT-5, Gemini-2.5-Pro, GLM, DeepSeek differ from from human evaluation. GPT-4.1 has the highest Spearman coefficient.
	- Different prompting strategies: `Simple Warning` prompting slightly helps models to solve problem variants.

**Strengths:**

- An attempt to quantify whether LLMs can overfit not only to some problems and problem types but to specific reasoning templates behind those problems.
- Highly curated set of tasks and 5 different variants of original problems.

**Weaknesses:**

- Considering striking improvement of OpenAI's GPT models, (from GPT-4o to GPT-5), in one generation LLMs can have substantial improvements (e.g. 8.89 to 71.43 on V2 variant), it's hard to imagine that the PROBE benchmark will be useful in the near future.
- Please see the questions section.

**Questions:**

- Why in the paradigm change when we change from "uneven" to "even" the original solution doesn't work? The rope would still take 30 min to burn if you light it from both ends and everything else still make sense. There's just another simpler solution but it doesn't mean that the original reasoning is wrong.
- Maybe a sampling strategy (e.g. majority voting) and/or other test-time scaling can be used to improve reasoning?
- What was  the language of questions? Was it Mandarin or English? In the supplementary material in the `questions.jsonl`, the data is in Mandarin which makes it hard to even assess the it. Also, there're no ground-truth solutions and models' responses which also makes it hard to reproduce and verify the core message of the paper.
- Prompting strategies: why there're no GPT-5 models in the table 5?
- The claim that since LLMs' performance drops on PROBE compared to the original problems means "reasoning overfitting" and lack of generalization cannot be taken seriously since humans (Tab. 2) show worse performance on V4 and V5 than for example GPT-5. Do humans also overfit then?


#### Notes

- Burning rope problem: it's better to say "variable rate of burning" instead of uneven since uneven makes the whole problem statement confusing.

---

### Official Review · Reviewer_QhVt · 2025-11-01

**Soundness:** 3
**Presentation:** 3
**Contribution:** 3
**Rating:** 4
**Confidence:** 5

**Summary:**

This paper presents a novel benchmark robust to reasoning overfitting. More specifically, PROBE uses problem variants that force a shift in the core reasoning paradigm. The measured performance of reasoning models achieve an accuracy of 81.57% on the original problems and drop up to 35.08% on variants.

The main claim is that models are predominantly effective at detecting simple forms of overfitting rooted in pattern matching, but inadequate for identifying more profound overfitting at the level of reasoning pathways. The proposed datasets consist of various types of puzzles with small variants to assess true reasoning, where variants are either simplification, unsolvability, and paradigm change.
To create such variations, humans are ask to rewrite a problem according to one of the variants. A second phase consists in verifying contradiction and meaningfulness of the problems. Stage 3 repeat another time stage 2 and measure inter-annotator agreement to filter even more the problems and keep the most correct ones.

In terms of experiments, the authors evaluate a various number of models using temperature of 0 and 16k tokens, and assess the final results with GPT-4.1. The authors also ask some human to solve the tasks. Without surprise, the results show a significant decrease in performance of all models, especially in unsolvability followed by simplification, while human performance is mostly above 90%. Interestingly certain models behave better with alternative (e.g. GPT5 vs Qwen).

The authors evaluate the quality of the LLM rater using their average scores on human answers to the problems (assuming humans are correct). I find this a flaky experiment and would strongly encourage the authors to conduct to replace the LLM-rater with human raters to assess the correctness of the produced results in Table 2. After that, the correlation with different LLM-raters can be done.

I do like the prompt experiment where different prompts are investigating, leading to different results. However, I am very curious why the top models of Table 2 are not investigated here (my hypothesis is that prompting differently may give similar results as the original problems). Additionally, I would like to see a simply chain-of-thoughts baseline (e.g., let's think step by step).

Finally, the authors evaluate pass@1 with a temperature set to 0. I do believe that it is unfair to evaluate models like that and using pass@k (k>=4) and other temperature would lead to a fairer comparison.

Overall, the paper is well structured and written, the motivation is sound. My main concerns are in the experiment design where human raters are not used (neither to compare with LLM-rater), prompting strategy are not investigated with stronger models, and most importantly, pass@1 is used with temperature=0 - please include higher pass@k.

**Strengths:**

- Benchmark to assess overfitting in reasoning
- Many models are compared

**Weaknesses:**

- LLM-Rater is not compared with human evaluator.
- Prompting experiments is not complete
- Measuring performance at pass@1 and temperature=0 is problematic.

**Questions:**

- Could you evaluate the produced answers by human raters and then compute correlations with LLM-Rater?
- Please include the results for the top-5 ranked models in Table 4.
- Could you report the results at pass@4 with more stochasticity?

---

### Official Review · Reviewer_icb6 · 2025-11-03

**Soundness:** 3
**Presentation:** 2
**Contribution:** 2
**Rating:** 2
**Confidence:** 4

**Summary:**

The paper introduces an evaluation framework (PROBE) developed to study what authors term reasoning-paradigm overfitting in large language models (LLMs). The authors argue that existing reasoning benchmarks are confounded by testing with already familiar, potentially memorized (eg by leakage into training) problems, unable to check for generalization to variants of already memorized templates. PROBE is meant to address this gap by introducing 5 controlled variant categories - termed simplification, unsolvability, paradigm change, numerical transformation and paraphrasing. Those are derived from 40 manually curated prototype problem templates, expanded into 216 total variants using the 5 categories.

Authors evaluate various LLMs, including closed API and open-weights models. The results indicate that depending on the model, there can be substantial performance drops from the original formulation on the variants. Strongest models like GPT-5 do not show strong fluctuations, while other models eg Kimi K2, can exhibit strong drops eg from 82.50\% on original to 28.89\% on V2. Human study with same problem variants provides human performance scores for comparison.

Authors further examine four prompting strategies: Straightforward, Simple Warning, Meta-Cognitive (CoT variant), and Role-Playing. Results suggest that drops in performance from the original persist across prompt types. Authors conclude from their observations that SOTA language and reasoning models overfit to a particular reasoning scheme and struggle with zero shot generalization when confronted with variants of problem that they can solve.

**Strengths:**

* Authors address important problem of testing generalization in SOTA LLMs and LRMs.
* Different problem variants reflect interesting types of possible reasoning scheme shifts.
* Using different prompt types ensures broad testing scenario.
* Authors test diverse SOTA LLMs and LRMs on recent frontiers.

**Weaknesses:**

1. Previous works, eg on AIW problems, https://arxiv.org/abs/2406.02061, show that already in simple problems, variations cause strong fluctuations in performance, even when variations do not change problem structure and difficulty at all. Authors here work is variations that change problem structure and difficulty. It is thus not a suprising finding given previous results to observe drops in performance (in AIW work, the drops are even stronger, despite the variations not changing problem structure and difficulty). It is for me somewhat hard to see how to argue for novelty with these observations that were already performed by others in simpler and more controlled settings.

2. Prompt type checking strategy. i) CoT like prompting in prompt types seems to me overly complicated and might affect the results. Given reasoning models are already employing thinking mode, a short "Think step by step and double check for errors before providing the final response" would have been good control compared to lengthy prompts employed in the study. ii) As Unsolvable is the category with the strongest drop, I miss a prompt that explicitly encourages models to check whether solution is impossible, eg via simple "Please provide an answer if correct solution exists; if none exists, state accordingly."

3. Details on human study are not provided (eg how many probands, time for response, etc)

4. No control for problem difficulty (it might be strong drops are induced only by problems that are itself more difficult in original formulation) or test time compute required for producing response (arguably, GPT-4o uses less than GPT-5, etc)

5. Evaluation was performed only with T=0 (greedy decoding, no sampling). This makes it impossible to estimate variance of correct responses given the input.

6. Missing relevant citations, eg https://arxiv.org/abs/2103.07191, https://arxiv.org/abs/2406.02061

**Questions:**

1. Would Unsolvable category be handled better if a prompt type would explicitly allow to state there is no solution like described above?
2. What would happen if using sampling with non zero temperature? Would the observed drops in performance remain similarly strong?
3. Why are the experiments with prompt types executed for only a subset of models? How was this subset selected?

---

### Note · Authors · 2025-12-03

I have read and agree with the venue's withdrawal policy on behalf of myself and my co-authors.